# Lymphocyte-to-C-Reactive Protein (LCR) Ratio Is Not Accurate to Predict Severity and Mortality in Patients with COVID-19 Admitted to the ED

**DOI:** 10.3390/ijms24065996

**Published:** 2023-03-22

**Authors:** Laure Abensur Vuillaume, François Lefebvre, Axel Benhamed, Amandine Schnee, Mathieu Hoffmann, Fernanda Godoy Falcao, Nathan Haber, Jonathan Sabah, Charles-Eric Lavoignet, Pierrick Le Borgne

**Affiliations:** 1Emergency Department, CHR Metz-Thionville, 57000 Metz, France; 2Department of Public Health, University Hospital of Strasbourg, 67000 Strasbourg, France; 3Service SAMU-Urgences, Centre Hospitalier Universitaire Édouard Herriot, Hospices Civils de Lyon, 69000 Lyon, France; 4Department of Gynecologic Surgery, Hôpitaux Universitaires de Strasbourg, 67000 Strasbourg, France; 5Emergency Department, Hôpital Nord Franche Comté, 90400 Belfort, France; 6Emergency Department, Hôpitaux Universitaires de Strasbourg, 67000 Strasbourg, France; pierrick.le-borgne@chru-strasbourg.fr; 7INSERM (French National Institute of Health and Medical Research), UMR 1260, Regenerative NanoMedicine (RNM), Fédération de Médecine Translationnelle (FMTS), 67000 Strasbourg, France

**Keywords:** lymphocytes, C reactive protein, LCR, COVID-19, severity, mortality

## Abstract

Health care systems worldwide have been battling the ongoing COVID-19 pandemic. Since the beginning of the COVID-19 pandemic, Lymphocytes and CRP have been reported as markers of interest. We chose to investigate the prognostic value of the LCR ratio as a marker of severity and mortality in COVID-19 infection. Between 1 March and 30 April 2020, we conducted a multicenter, retrospective cohort study of patients with moderate and severe coronavirus disease 19 (COVID-19), all of whom were hospitalized after being admitted to the Emergency Department (ED). We conducted our study in six major hospitals of northeast France, one of the outbreak’s epicenters in Europe. A total of 1035 patients with COVID-19 were included in our study. Around three-quarters of them (76.2%) presented a moderate form of the disease, while the remaining quarter (23.8%) presented a severe form requiring admission to the ICU. At ED admission, the median LCR was significantly lower in the group presenting severe disease compared to that with moderate disease (versus 6.24 (3.24–12) versus 12.63 ((6.05–31.67)), *p* < 0.001). However, LCR was neither associated with disease severity (OR: 0.99, CI 95% (0.99–1)), *p* = 0.476) nor mortality (OR: 0.99, CI 95% (0.99–1)). In the ED, LCR, although modest, with a threshold of 12.63, was a predictive marker for severe forms of COVID-19.

## 1. Introduction

For almost three years now, medical systems worldwide have been battling an ongoing pandemic caused by a novel coronavirus, SARS-CoV-2 (severe acute respiratory syndrome coronavirus 2). According to the World Health Organization, as of January 2023, three years after the start of the pandemic, this emerging virus infected almost 700 million people and resulted in nearly 7 million deaths around the world. In this global health crisis, Emergency Departments (ED) have been fighting at the front lines. As with other acute pathologies (such as trauma patients, stroke and sepsis), accurate triage is crucial. It implies rapid identification of the most critical patients in order to optimize patient management [1].

Complete blood count (CBC) is an easily accessible and inexpensive routine set of medical laboratory tests. Numerous studies have described changes in white blood cell counts of patients with COVID-19, including a significant decrease in circulating lymphocytes [2,3,4]. Similarly, some other studies suggest that C Reactive Protein (CRP) level may predict the risk of COVID-19 aggravation [5,6]. The Lymphocyte to C Reactive Protein Ratio (LCR) is the ratio between the absolute number of lymphocytes and the CRP level. It has already been studied as a predictive factor of the negative evolution of the oncological population, mainly with patients affected from digestive cancers. It has also been involved in prognostic scores [2,7,8]. Thus, it is recognized as an excellent ratio used in colon cancer [9]. Systemic inflammation via host–tumor interactions is currently recognized as a characteristic of cancer and is one of the indicators of tumor progression. LCR thus seems to be a good reflection of complex immunological interactions between the host and the tumor, which lead to a systemic inflammatory process, and would contribute to the pathogenesis and progression of carcinomas. Indeed, during the systemic inflammation phase, CRP is synthesized by the liver under the impulse of interleukin-6 (IL6) as soon as a tissue injury is detected by macrophages [10]. CRP, via IL6, also recruits other cells by chemotaxis, in particular lymphocytes [11]. Thus, unregulated and large-scale production of interleukins leads to an increase in CRP and a mobilization of lymphocytes [12].

With inflammation in COVID-19 being important and related to the severe manifestations of the infection, we observe a “relative” lymphopenia (balanced between lymphocyte stimulation and viral destruction) correlated to the disease’s severity [13,14,15]. However, in viral infection, which is also the target of inflammation, there is generally a progressive decrease in lymphocytes, which, once recruited, are targets for the viruses. LCR could thus have an interesting prognostic value in COVID-19, just as in cancer but in reversed proportions, inflammation being one of the key factors of the severity of COVID-19. However, there is little published information on LCR in COVID-19 patients and the evidence should be strengthened [16,17,18,19]. In addition, the cohorts studying LCR were small and may have had confounding factors [16,17,18]. These confounding factors include history or treatments that may alter the lymphocyte count (e.g., chemotherapy, immunosuppressive therapy, long- and short-term corticosteroid therapies, pre-admission antibiotic therapy, active cancer or hematological malignancies). We hypothesize that LCR is less efficient in a selected population.

This study aimed to assess the prognostic value of LCR in the severity and mortality of patients infected with SARS-CoV-2 on admission to the ED in a cohort selected to limit confounding factors. The secondary objective was to determine the LCR threshold for which an increased risk of severe disease is predicted. 

## 2. Results

### 2.1. Characteristics of the Study Population

During the study period, a total of 49,326 patients were admitted to the ED of all six hospitals. Of these patients, 4470 had a laboratory-confirmed SARS-CoV-2 infection and, in total, 1035 patients were included in our study.

Our cohort had a median age of 69 (58–79) years and was predominately male (58.8%, 95% CI: (55.8;61.8)). One-third of the study population was obese (33%). In terms of medical history, over half of the patients (56.7%) had high blood pressure, over a quarter of them (26.7%) had a history of diabetes, and 23.2% of them presented pre-existing renal failure. More than three-quarters of the patients (77.2%, CI95%: 74.6–79.8%) did not show any loss of functional autonomy, as measured by the Knaus score.

The vast majority of patients (92.8%, CI95%: 91.1–94.3%) presented lymphopenia (below 1000/mm^3^) upon admission to the ED, and this decrease in lymphocytes was significantly higher in severe patients (0.9 vs. 0.78, *p* = 0.003). Similarly, CRP was also higher in patients with severe COVID-19 (68 vs. 124 mg/L, *p* < 0.001).

At ED admission, the median LCR was significantly lower in the group presenting severe disease compared to that with moderate disease (6.24 (3.24–12) versus 12.63 (6.05–31.67), *p* < 0.001). Principal clinical and biochemical patient characteristics are summarized in Table 1.

### 2.2. Biochemichal Factors Associated with Severe COVID-19

Of the total study population, 789 patients (76.2%) had moderate disease, whereas 246 (23.8%) had severe disease, which required ICU management. In univariate analysis, lymphocyte were not associated and CRP and LCR were associated. Therefore, in multivariate analysis, the factors associated with the severity of the infection were CRP (OR: 1.009, 95% CI: (1.006–1.011), *p* < 0.001); LCR was not associated (OR: 0.99, 95% CI (0.99–1), *p* = 0.476). These values are summarized in Table 2.

### 2.3. Predictive Factors of Severe COVID-19

We determined two ROC curves to predict the risk of disease severity. Regarding LCR at ED admission, the area under the curve (AUC) was 0.684 (95% CI: (0.646–0.721), *p* < 0.001). The best cutoff for predicting the risk of infection severity was 12.79; it yielded a sensitivity of 78.51% (95% CI (72.8–0.83)) and a specificity of 49.68% (95% CI (46.1–52.3)). These results are presented in Figure 1.

### 2.4. Biochemical Factors Associated with Mortality

A total of 139 patients died during their hospital stay, representing 13.4% (95% CI: (11.4–15.5)) of our cohort. Upon admission to the ED and at H-24, the LCR values were not associated with mortality in univariate analysis 0.998 (95% CI: (0.995–1.001) *p*: 0.253) and not associated in multivariate analysis 0.996 (95% CI (0.988–1.004) *p*: 0.354). However, no parameter was significantly associated with mortality in multivariate analysis. These results are summarized in Table 3.

### 2.5. Predictive Factors of Mortality

Regarding LCR at admission, the AUC was 60.71 (55.46; 65.96) *p* < 0.001.The best LCR threshold for predicting the risk of death was 6.27; it yielded a sensitivity of 48.12% (95% CI: (39.4–57)) and a specificity of 70.8% (95% CI: (67.6–73.8)). These results are presented in Figure 2.

In univariate analysis, if LCR was lower than 12.79, the OR was valued significant with severe COVID-19 and mortality with, respectively, OR 3.607 (95%CI (2.574–5.055) *p* < 0.001) and OR 1.406 (95%CI (1.158–2.512) *p*: 0.007). However, in multivariate analysis, LCR is just associated with severe disease, if LCR was lower than 12.79. The OR was valued significant at 2.025 (95% CI: (1.18–3.48), *p* = 0.011) with severe disease and not significant OR 1.160 (CI95% (0.571–2.356) *p*: 0.681) with mortality; with propensity score analysis, severe disease was associated with LCR (OR:2.95 (2.002–4.363) *p* < 0.001). All result are presented in Table 4.

## 3. Discussion

Our study included a retrospective multicenter cohort of patients with COVID-19 and was selected to limit bias on the interpretation of lymphocyte counts. Our main objective was to study the prognostic value of LCR in patients admitted to the ED and then all patients hospitalized for SARS-CoV-2 infection. We showed that LCR remained a modest marker of severity of SARS-CoV-2 infection, with a critical threshold of 12.8.

Our study presents more moderate results than those found in the literature. Firstly, Ullah et al., who also found superiority to NLR, in a cohort of 176 patients with COVID-19, showed that low LCR on admission predicts the need for intensive care (1.011 vs. 632, *p* = 0.04) and is a prognostic score for overall mortality (OR 0.2, 0.06–0.47, *p* = 0.001), with a threshold of 100 [17]. Furthermore, Albarrán-Sánchez et al., in a cohort of 242 patients in Mexico, showed that a lower LCR (0.03 vs. 0.06, *p* < 0.002) was predictive of mortality [18]. However, the common point of these studies was less restrictive inclusion criteria than ours, which we believe is an asset of our study.

Indeed, in order to study in a rigorous way the influence of a health condition on the complete blood count, it is important to select patients. Studies on COVID-19 have often been retrospective, due to the high pressure on the health care system, but also due to the rapid evolution of the virus. Most studies have focused on the inclusion of patients with SARS-CoV-2 infection, without necessarily taking into account confounding factors. These confounding factors are important in epidemiology. These studies, in the absence of prospective trials that would not have been relevant, have nevertheless provided additional knowledge about COVID-19. In epidemiology, association marks a link with causality. Causality is more likely when there is a dose–response relationship and confounding factors have been properly accounted for [20]. Taking up the grid for judging causality proposed by Hill [21], there is a pathophysiological plausibility to the association found between CSF and the severity of COVID-19; we have adjusted for a number of confounding factors, but we cannot affirm that there are no confounding factors not taken into account. Thus, we cannot affirm causality formally, but simply describe the association, which seems to be lesser, however, when we rigorously select our cohort.

The LCR in the context of COVID-19 thus remains globally less efficient than in oncology patients. Even if the inflammation remains common between the two pathologies, there seem to be pathophysiological differences. SARS-CoV-2 is a virus that uses angiotensin converting enzyme 2 (ACE2) as its primary cellular receptor to enter the host cell [22]. After an incubation period, patients develop polymorphic respiratory and/or digestive symptoms. Infection of the epithelial and immune cells of the respiratory tract generates danger signals, which are recognized by different receptors that bind to viral RNA or viral surface proteins. These receptors will then activate transcription factors leading to the secretion of cytokines (TNF-α, IL-1, IL-6). These lead to capillary hyperpermeability and the attraction of inflammatory cells, and type I interferons (IFN-1), which promote the expression of target genes [23]. The type I interferon pathway inhibits viral replication, protects non-infected cells and stimulates antiviral lymphocyte immunity (CD8 T cells, NK cells) leading to lysis of infected cells [24]. Activation of transcription factors leads to an initial cytokine secretion by infected cells, which allows viral lysis and ensures the establishment of a lasting immunity. This phase of viral invasion is followed, in some patients, by an inadequate host immune response marked by worsening of respiratory symptoms and inflammatory syndrome, usually eight to ten days after the first symptoms [25]. This dysimmune phase is sometimes called cytokine storm, and is assimilated to a true viral “sepsis” [14]. It results from the inefficiency of the initial immune response, which leads to an amplification of the inflammatory response responsible for the clinical picture [26]. Thus, high levels of cytokines, including circulating interleukins, have been reported in patients with severe COVID-19 [25]. For these and other reasons, some researchers have focused on immunomodulators and immunosuppressants, including corticosteroids and remedivir, to reduce the effects of severe COVID. However, the only weapon to date to prevent severe forms remains vaccination

Lastly, and retrospectively, we could say that one of the major cornerstones in the management of this outbreak was controlling the number of patients in the ICU, which was the main reason behind the oversaturation of health care systems, an overload that increases mortality [27]. Therefore, the accurate orientation of a patient with COVID-19 in the ED between a conventional hospitalization unit or the ICU is essential.

Other hematological ratios have been studied such as PLR, NLR or SSI since the beginning of the pandemic [28,29]. The value of the LCR should probably be coupled with other markers in a multi-marker approach or in a combined severity prediction score; this might be the key to allowing optimization and standardization of patient triage in the ED.

### Limitations

Our study has several limitations. First, our study was retrospective. However, even today, there are very few prospective studies on COVID-19. Secondly, we had to exclude an extensive number of patients due to their medical history, such as onco-hematological or immunocompromised patients, as it modifies blood counts and, therefore, circulating lymphocytes rate. Furthermore, we did not collect CRP at 24 h, which did not allow us to study the early variation of this ratio, which could have been interesting. Thirdly, due to the absence of a control group (suspected COVID), we were unable to study the diagnostic performance of LCR in SARS-CoV-2 infection. Further studies are required, notably on larger prospective cohorts, and probably as part of a multimarker approach. Fourthly, this study remains based on a COVID-19 cohort of the alpha variant, but still provides strong evidence on COVID-19 pathophysiology. Thus, our cohort study patients remain “totally” naive, notably to vaccination and treatments (such as steroids), with a quite virulent variant. Lastly, given the overwhelming workload and pressure on our health care systems during the first wave of the pandemic, overall parameters of the hospital stay could not be detailed.

## 4. Materials and Methods

### 4.1. Study Population and Settings

A retrospective multicenter study was conducted in six ED of the Northeast region of France. We directed the study in two university-affiliated hospitals (CHRU of Strasbourg and CHU of Reims) and four general hospitals (Colmar Hospital, Nord Franche-Comté Hospital, Metz-Thionville Hospital and Haguenau Hospital). These health care facilities, along with the entire Greater-East region of France, constituted one of the outbreak’s epicenters in Europe during the first COVID-19 wave. As of the end of 2022, this area reported nearly 15,000 deaths and 78,000 hospitalized patients infected with SARS-CoV-2.

We have included all adult patients who were hospitalized with a diagnostic of COVID-19 infection after presenting (positive RT-PCR) to the ED between 1 March and 30 April 2020. All patients included in our study had a laboratory-confirmed diagnosis of COVID-19 by RT-PCR on nasopharyngeal swab. We excluded patients who had a non-confirmed diagnosis, those who received outpatient care, and those who received palliative therapy or limitation of therapeutic effort upon admission to the ED. Patients with a medical history or treatment that altered their blood counts and, therefore, their circulating lymphocytes or neutrophils (e.g., chemotherapy, immunosuppressive therapy, long- and short-term corticosteroid therapy, pre-admission antibiotic therapy, active cancer, or hematological malignancies) were also excluded.

### 4.2. Data Collection

We retrospectively queried patients’ electronic medical records for epidemiological, clinical, and biochemical data and then standardized the results in a report file. We recorded symptom onset date along with patient’s current treatment and medical history (including cardiovascular comorbidities, diabetes, pre-existing renal failure, cancer, and hematological diseases). The primary endpoint was the prognostic value of LCR on in-hospital mortality. The secondary endpoint was its prognostic value on severity of disease, where severe disease was defined by patient admission to the ICU (patients under invasive mechanical ventilation), and moderate disease was defined by patient admission to conventional hospitalization units (most patients with oxygen therapy). Ambulatory patients were excluded. Obesity was defined by a body mass index superior to 30 kg/m^2^. Functional autonomy was measured by the Knaus score. Standard biochemical parameters, such as levels of creatinine, CRP, total leukocytes and lymphocytes, were also collected. All collected data are summarized in the results sections.

### 4.3. Ethics

This study was approved by the local ethics committee of the University of Strasbourg in France (reference CE: 2020-39), which, in accordance with the French legislation, waived the need for informed consent of patients whose data were entirely retrospectively studied.

### 4.4. Statistical Analysis

The statistical analysis consisted of two parts: a descriptive and an inferential analysis. For the descriptive statistical analysis, the frequencies and the proportions of the categorical variables and the median and the first and third quartiles of each continuous variable were given. To compare the continuous covariates, Wilcoxon tests were performed and to compare the categorical covariates, Chi-Squared tests or Fisher tests were performed. Multivariate analyses were performed using all significant variables in univariate analysis and relevant variables that could explain severity or mortality [30]. A propensity score with an LCR greater than a threshold was calculated and weighted logistic models to predict severity or mortality were carried out with the overlap method. ROC curves to predict the best thresholds of the biological parameters to predict infection with the Youden method were performed. Analyses were performed with R software (version 4.1.2), as well as with all the software packages required to carry out the analysis.

## 5. Conclusions

LCR, although modest, was a predictive marker for severe forms of COVID-19 at ED admission. It could be useful to identify patients at risk of poor outcome and this may be relevant in combination with all other known markers of severity for the disease.

## Figures and Tables

**Figure 1 ijms-24-05996-f001:**
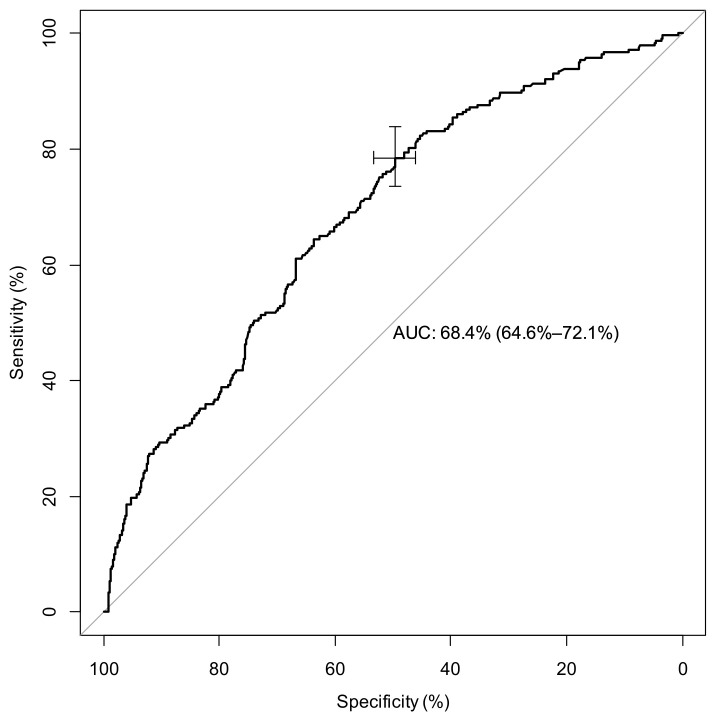
ROC curve for LCR found at admission, a predictive ratio to determine severe COVID (Moderate vs. Severe COVID = ICU admission). Legend: AUC= Area under curve.

**Figure 2 ijms-24-05996-f002:**
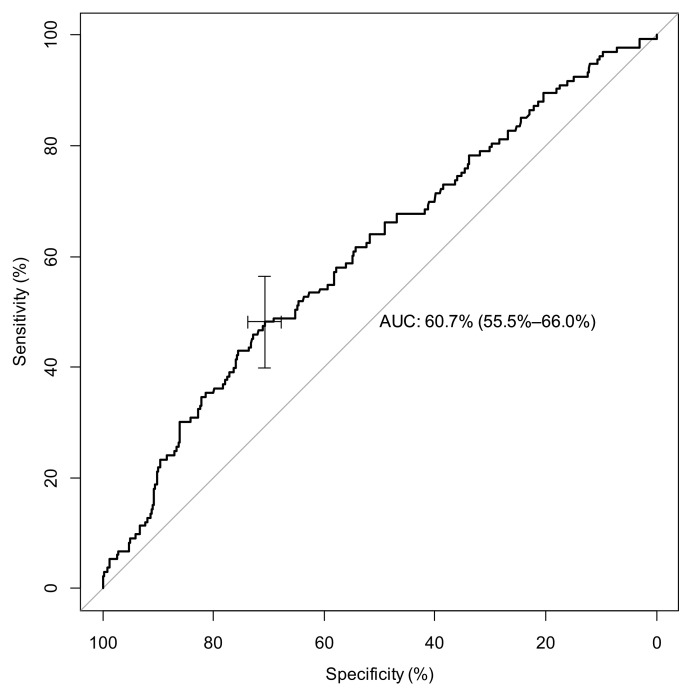
ROC Curve for admission LCR, predictive ratio to determine severe COVID cases (surviving vs. deceased patients). Legend: AUC= area under curve.

**Table 1 ijms-24-05996-t001:** Demographic, baseline, and laboratory characteristics of patients with COVID-19.

		All Patients (n = 1035)	Moderate COVID-19 (n = 789)	Severe COVID-19 (n = 246)	*p* Value
Age (years)		69.0 (58.0; 79.0)	70.0 (58.0; 81.0)	66.0 (57.3; 72.0)	<0.001 *
Male (%)		609 (58.8)	433 (54.9)	176 (71.5)	<0.001 *
Smokers		46 (4.4)	34 (4.3)	12 (4.9)	0.706
**Comorbidities**					
Hypertension		587 (56.7)	453 (57.4)	134 (54.5)	0.416
Diabetes		275 (26.6)	202 (25.6)	73 (29.7)	0.207
Obesity	BMI (30; 40) (kg/m^2^)	253 (33.2)	172 (31.2)	81 (38.6)	0.056
	≥40 (kg/m^2^)	28 (3.7)	21 (3.8)	7 (3.3)	0.966
COPD		56 (5.4)	44 (5.6)	12 (4.9)	0.672
Pre-existing renal failure		237 (23.2)	199 (25.5)	38 (15.8)	0.002 *
Cardiovascular disease		357 (34.5)	291 (36.9)	66 (26.8)	0.004 *
**Lab results**					
Neutrophiles, ×10^9^ per L		4.930 (3.430; 6.932)	4.730 (3.370; 6.620)	5.510 (3.760; 8.160)	<0.001 *
Lymphocytes, ×10^9^ per L		0.870 (0.630; 1.200)	0.900 (0.640; 1.220)	0.780 (0.590; 1.122)	0.003 *
Platelets, ×10^9^ per L		194.5 (152.0; 248.0)	196.0 (154.0; 247.0)	192.0 (144.0; 253.0)	0.518
CRP, mg/L		81.0 (39.0; 142.3)	68.0 (33.0; 128.0)	124.0 (76.0; 192.0)	<0.001 *
LCR		10.35 (5.28; 25.53)	12.63 (6.05; 31.67)	6.24 (3.32; 12.0)	<0.001 *
Mortality n (%)		139 (13.6)	82 (10.4)	57 (24.1)	<0.001 *
Hospitalization duration (days)		10.0 (7.0; 17.3)	8.0(6.0; 12.0)	24.0(17.0; 38.0)	<0.001 *

Data are expressed in median (Q1–Q3) n (%), where n is the total number of patients with available data. * *p* < 0.05. Abbreviations: BMI = body mass index, CKD = Chronic kidney disease, COPD = chronic obstructive pulmonary disease CRP = C reactive protein, LCR = Lymphocyte to CRP ratio.

**Table 2 ijms-24-05996-t002:** Associated factors linked to severe COVID-19.

	Moderate COVID	Severe COVID	Univariate Analysis		Multivariate Analysis	
			OR IC 95%	*p*	OR IC 95%	*p*
Lymphocytes 109/L	0.900 (0.640; 1.220)	0.780 (0.590; 1.122)	0.827 (0.616; 1.110)	0.206	1.154 (0.825; 1.613)	0.403
CRP, mg/L	68.0 (33.0; 128.0)	124.0 (76.0; 192.0)	1.009 (1.007; 1.011)	<0.001	1.009 (1.006; 1.011)	<0.001 *
LCR	12.63 (6.05; 31.67)	6.24 (3.32; 12.0)	0.990 (0.985; 0.996)	<0.001	0.999 (0.995; 1.002)	0.476

Legend: CRP = C reactive Protein; LCR = Lymphocyte to CRP Ratio, * = *p* < 0.005.

**Table 3 ijms-24-05996-t003:** LCR and in-hospital mortality.

	Surviving Patient	Deceased Patient	Univariate Analysis		Multivariate Analysis	
			OR IC 95%	*p*	OR IC 95%	*p*
Lymphocytes 109/L	0.890 (0.650; 1.220)	0.720 (0.500.; 1.000)	0.524 (0.336; 0.815)	0.004 *	0.739 (0.403; 1.358)	0.330
CRP (mg/L)	78.5 (37.0; 139.0)	100.0 (56.0; 158.0)	1.003 (1.001; 1.005)	0.006	1.001 (0.997; 1.004)	0.599
LCR	11.05 (5.64; 27.81)	7.33 (3.85; 18.37)	0.998 (0.995; 1.001)	0.253	0.996 (0.988; 1.004)	0.354

Legend: CRP = C reactive Protein; LCR = Lymphocyte to CRP Ratio, * = *p* < 0.005.

**Table 4 ijms-24-05996-t004:** Predictive value of LCR with threshold of 12.79.

	≥Threshold ROC 12.79 n (%)	<Threshold ROC 12.79 n (%)	Univariate Analysis		Multivariate Analysis		Propensity Score Analysis	
			OR IC 95%	*p*	OR IC 95%	*p*	OR IC 95%	*p*
Severe COVID-19	52 (11.9)	190 (32.8)	3.607 (2.574; 5.055)	<0.001 *	2.025 (1.179; 3.479)	0.011 *	2.955 (2.002; 4.363)	<0.001 *
Moderate COVID-19	385 (88.1)	390 (67.2)						
Deceased Patient	43 (9.9)	90 (15.8)	1.406 (1.158; 2.512)	0.007 *	1.160 (0.571; 2.356)	0.681	1.133 (0.677; 1.897)	0.635
Surviving Patient	392 (90.1)	481 (84.2)						

Legend: * = *p* < 0.05.

## Data Availability

Not applicable.

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
