# Peer review of "Lymphocyte-to-C-Reactive Protein (LCR) Ratio Is Not Accurate to Predict Severity and Mortality in Patients with COVID-19 Admitted to the ED"

_ijms, 2023, doi:10.3390/ijms24065996_

Round 1
Reviewer 1 Report
Good morning
for the Authors,
Analyzing with attention and interest the Manuscript (Article) with ID: ijms-2264435-peer-review-v1, entitled "Lymphocyte-to-C-reactive protein (LCR) ratio is not accurate to predict severity and mortality in COVID-19 patients admitted to the ED" for a possible publication in Journal International Journal of Molecular Sciences – MDPI (ISSN:1422-0067; IF=6.208).
In Conclusion:
Reconsider after major revision of this manuscript!

Author Response
#Reviewer1
Open Review
(x) I would not like to sign my review report
( ) I would like to sign my review report
Quality of English Language
( ) English very difficult to understand/incomprehensible
( ) Extensive editing of English language and style required
(x) Moderate English changes required
( ) English language and style are fine/minor spell check required
( ) I am not qualified to assess the quality of English in this paper
Yes |
Can be improved |
Must be improved |
Not applicable |
|
Does the introduction provide sufficient background and include all relevant references? |
( ) |
(x) |
( ) |
( ) |
Are all the cited references relevant to the research? |
( ) |
(x) |
( ) |
( ) |
Is the research design appropriate? |
( ) |
(x) |
( ) |
( ) |
Are the methods adequately described? |
( ) |
(x) |
( ) |
( ) |
Are the results clearly presented? |
( ) |
( ) |
(x) |
( ) |
Are the conclusions supported by the results? |
( ) |
(x) |
( ) |
( ) |
Comments and Suggestions for Authors
Good morning
for the Authors,
Analyzing with attention and interest the Manuscript (Article) with ID: ijms-2264435-peer-review-v1, entitled "Lymphocyte-to-C-reactive protein (LCR) ratio is not accurate to predict severity and mortality in COVID-19 patients admitted to the ED" for a possible publication in Journal International Journal of Molecular Sciences – MDPI (ISSN:1422- 0067; IF=6.208).I consider that:The authors of the article have proposed a much-discussed topic in the medical scientific world today, namely: the levels blood of the inflammatory markers present in COVID-19.
1. In Chapter 1 – Introduction (well structured) the authors presented the reasons for
choosing their study which are congruent with those of other authors cited as references.
2. In Chapter 2 – Methods:
- The authors clearly presented the inclusion and exclusion criteria of patients in their
study.
- The authors presented the consent of the ethics committee of the institutions where
they conducted the study, in accordance with the legislation.
- To process the obtained data, the authors used different tests and specific statistical
analysis coefficients (p and r value), Chi-Squared tests or Fisher tests, ROC curves,
using a powerful R software (version 4.1.2). They had a significant value in assessing
the friability of the study.
3. In Chapter 3 – Results:
- Table1: Check the statistical data marked in yellow!
- What is the abbreviation of COPD?
Response: Thanks to the reviewer for pointing out this oversight. COPD= chronic obstructive pulmonary disease. We have added it in the legend.
The percentages in brackets are correct: they are higher than expected because only non-missing data are taken into account.
- Line 160-161 the authors specify: ’’Abbreviations: BMI=body mass index,
CKD=Chronic kidney disease, ED=Emergency Department, O2=oxygen,
°C=Celsius degree...’’ abbreviations that do not appear presented in Table 1!
Response: we made changes.
- Line 188 the authors specify: ’’A total of 139 patients died during their hospital stay,
representing 13.4%...”
- Out of 1035 total patients, so it means that n=896 survived!
- BUT Table 1 and Table 4 they do not match!
- The statistical data in Tables 2, 3 and 4 are difficult to identify/compare in Figures 2
and 3!
Response:
There are 12 missing data concerning mortality and 16 concerning LCR.
- In Chapter 4 - Discussion: The authors compared their results with those of other
authors according to the bibliography/references.
5. All authors had a fair contribution in the realization of the study.
6. The bibliography chosen by the authors corresponds to the requirements and refers to the subject of the article. In Conclusion: Reconsider after major revision of this manuscript!
We would like to thank the reviewer 1 for his help and his suggestions to improve ou work.

Reviewer 2 Report
I read it with great interest, but I have raised several concerns.
#1. Please use the patient first language.
COVID-19 patients -> patients with COVID-19
#2. Please add the hypothesis of this study.
#3. Multi- 133 variate analyses were performed using all significant variables in univariate analysis and 134 relevant variables that could explain severity or mortality. -> Please cite the statistical guideline below.
DOI: https://doi.org/10.54724/lc.2022.e3
#4. I think PS mathcing is needed. DOI: https://doi.org/10.54724/lc.2022.e18
#5. This is an excellent paper.
Author Response
#Reviewer2
Open Review
( ) I would not like to sign my review report
(x) I would like to sign my review report
Quality of English Language
( ) English very difficult to understand/incomprehensible
( ) Extensive editing of English language and style required
( ) Moderate English changes required
(x) English language and style are fine/minor spell check required
( ) I am not qualified to assess the quality of English in this paper
Yes |
Can be improved |
Must be improved |
Not applicable |
|
Does the introduction provide sufficient background and include all relevant references? |
(x) |
( ) |
( ) |
( ) |
Are all the cited references relevant to the research? |
(x) |
( ) |
( ) |
( ) |
Is the research design appropriate? |
(x) |
( ) |
( ) |
( ) |
Are the methods adequately described? |
(x) |
( ) |
( ) |
( ) |
Are the results clearly presented? |
(x) |
( ) |
( ) |
( ) |
Are the conclusions supported by the results? |
(x) |
( ) |
( ) |
( ) |
Comments and Suggestions for Authors
I read it with great interest, but I have raised several concerns.
#1. Please use the patient first language.
COVID-19 patients -> patients with COVID-19
Response: We made changes.
#2. Please add the hypothesis of this study.
Response: We made changes.
#3. Multi- 133 variate analyses were performed using all significant variables in univariate analysis and 134 relevant variables that could explain severity or mortality. -> Please cite the statistical guideline below.
DOI: https://doi.org/10.54724/lc.2022.e3
Response: Done!
#4. I think PS matching is needed. DOI: https://doi.org/10.54724/lc.2022.e18
Response: A propensity score was used to weight logistic models. We preferred propensity score weighting to matching in order not to lose subjects.
#5. This is an excellent paper.
We would like to thank the reviewer 2 for his help and his suggestions to improve ou work.

Reviewer 3 Report
This is an interesting study that fits the Journal scope who evaluate the predictive role of Lymphocyte-to-C-reactive protein (LCR) ratio in severity and mortality in COVID-19 patients admitted to the ED.
The authors concluded that LCR isa modest predictive marker for severe forms of COVID-19, but not for mortality. Overall, the manuscript is well written.
I’m very curios, if the authors provide the total lymphocytes, neutrophiles, and platelets number, why they don’t introduce the hematological ratios (NLR, PLR, and SII) in the analyze. Based on the recently published paper in the literature, the abovementioned markers are predictors of mortality, ICU admission, severity of invasive mechanical ventilation. See the following article:
- https://doi.org/10.3390/diagnostics12112757
- https://doi.org/10.3390/microorganisms11020319
- https://doi.org/10.3390/diagnostics12102379
- https://doi.org/10.3390/diagnostics12092089
- https://doi.org/10.3390/diagnostics13040746
- https://doi.org/10.3390/healthcare11030387
- https://doi.org/10.3390/vaccines10081233
Moreover, I suggest introducing the abovementioned markers in the results section and to compare the predictive role of this inflammatory ratios with the one that authors presented. Feel free to add the abovementioned references in the discussion section and compare your results with the others articles results.
Author Response
#Reviewer3
Open Review
(x) I would not like to sign my review report
( ) I would like to sign my review report
Quality of English Language
( ) English very difficult to understand/incomprehensible
( ) Extensive editing of English language and style required
( ) Moderate English changes required
(x) English language and style are fine/minor spell check required
( ) I am not qualified to assess the quality of English in this paper
Yes |
Can be improved |
Must be improved |
Not applicable |
|
Does the introduction provide sufficient background and include all relevant references? |
( ) |
(x) |
( ) |
( ) |
Are all the cited references relevant to the research? |
( ) |
( ) |
(x) |
( ) |
Is the research design appropriate? |
( ) |
( ) |
(x) |
( ) |
Are the methods adequately described? |
( ) |
(x) |
( ) |
( ) |
Are the results clearly presented? |
( ) |
( ) |
(x) |
( ) |
Are the conclusions supported by the results? |
( ) |
(x) |
( ) |
( ) |
Comments and Suggestions for Authors
This is an interesting study that fits the Journal scope who evaluate the predictive role of Lymphocyte-to-C-reactive protein (LCR) ratio in severity and mortality in COVID-19 patients admitted to the ED.
The authors concluded that LCR isa modest predictive marker for severe forms of COVID-19, but not for mortality. Overall, the manuscript is well written.
I’m very curios, if the authors provide the total lymphocytes, neutrophiles, and platelets number, why they don’t introduce the hematological ratios (NLR, PLR, and SII) in the analyze. Based on the recently published paper in the literature, the abovementioned markers are predictors of mortality, ICU admission, severity of invasive mechanical ventilation. See the following article:
- https://doi.org/10.3390/diagnostics12112757
- https://doi.org/10.3390/microorganisms11020319
- https://doi.org/10.3390/diagnostics12102379
- https://doi.org/10.3390/diagnostics12092089
- https://doi.org/10.3390/diagnostics13040746
- https://doi.org/10.3390/healthcare11030387
- https://doi.org/10.3390/vaccines10081233
Response :
Indeed, we did collect every white blood cells and most biomarkers available in the ED count in our cohort. We chose to focus only on this ratio (LCR) in this study. Firstly, our cohort study patients remain “totally” naïve notably to vaccination and treatments (such as steroids) and we excluded an extensive number of patients due to their medical history, such as onco-hematological or immunocompromised patients, as it modifies CBC. That makes our results very interesting in terms of validity.
Secondly, We chose to focus on this ratio mostly because of the two parts, lymphopenia (well studied in COVID-19 patients) and C-reactive protein (inflammatory biomarker routinely used in the ED).
Moreover, I suggest introducing the abovementioned markers in the results section and to compare the predictive role of these inflammatory ratios with the one that authors presented. Feel free to add the abovementioned references in the discussion section and compare your results with the others articles results.
Response: Finally, you are absolutely right, there are other ratios (NLR, PLR, SSI) and plenty of articles published about them. We added to our manuscript some of those informations and references in the discussion section. But we chose, because we have already plenty of datas in the results section and we would like to stay readable. As you probably know the perfect biomarker does not exist, we probably need to use a multimarker approach and built a clinic-biological score in order to improve our results to predict the severity/mortality on COVID-19 patients in the ED.
We would like to thank the reviewer 3 for his help and his suggestions to improve ou work.

Round 2
Reviewer 1 Report
Good morning for all authors,
RE-Analyzing with attention and interest the Manuscript (Article) with ID: ijms-2264435-peer-review-v2, entitled "Lymphocyte-to-C-reactive protein (LCR) ratio is not accurate to predict severity and mortality in patients with COVID-19 admitted to the ED" for a possible publication in Journal International Journal of Molecular Sciences – MDPI (ISSN:1422-0067; IF=6.208);
I consider that:
1. The article follows all the specific instructions of the journal presented in: aims and scope, instructions for authors and other information about the journal.
2. In Chapter – Results: the data presented in this manuscript are well structured and coherent.
In conclusion:
I Accept in present form ijms-2264435-peer-review-v2!
Reviewer 2 Report
I have no further comments.